# Parental Psychological Control and Addiction Behaviors in Smartphone and Internet: The Mediating Role of Shyness among Adolescents

**DOI:** 10.3390/ijerph192416702

**Published:** 2022-12-13

**Authors:** Qi Zhang, Guangming Ran, Jing Ren

**Affiliations:** 1College of Preschool and Primary Education, China West Normal University, Nanchong 637002, China; 2Department of Psychology, Institute of Education, China West Normal University, Nanchong 637002, China

**Keywords:** parental psychological control, shyness, smartphone addiction, internet addiction, adolescents

## Abstract

Parental psychological control has been found to be a vital familial factor that is closely related to adolescents’ addiction behaviors with regard to smartphones and the internet. However, the underlying mechanisms of these associations are less clear. The aim of the present study was to examine whether shyness mediated the relationships between parental psychological control and these two addiction behaviors. A positivist paradigm was used in the present study. The questionnaires (parental psychological control, shyness, and smartphone and internet addiction questionnaires) were used to collect data from a sample of 1857 Chinese adolescents (961 female, 896 male) in junior and senior middle schools. Descriptive statistics as well as correlation and mediation tests were employed to analyze the data. We observed that adolescents with siblings showed a higher level of internet addiction than those with no siblings. Moreover, three dimensions of parental psychological control were positively associated with addiction behaviors. The following analyses displayed that the correlation between authority assertion and smartphone addiction was greater than that between authority assertion and internet addiction. Subsequently, shyness was significantly positively related to parental psychological control and addiction behaviors. Importantly, we found that the relations between the three dimensions of parental psychological control and the addiction behaviors concerning smartphones and the internet were partially mediated by shyness. This study contributes to our understanding of how parental psychological control predicts high levels of adolescents’ addiction behaviors surrounding smartphones and the internet.

## 1. Introduction

With the growing popularity of smartphone and internet usage among adolescents, there is concern that their excessive use of these two technologies (adolescents’ addiction behaviors surrounding smartphones and the internet) are serious risk factors for the development of psychological and physical difficulties in adulthood [1,2,3]. These addiction behaviors in adolescents have become public health problems in recent years, leading to serious consequences for families and society [3]. Despite that smartphone and internet addiction is not specifically acknowledged in the fifth edition of the Diagnostic and Statistical Manual of Mental Disorders [4], a growing number of researchers tend to regard them as addiction behaviors [3,5,6]. Accordingly, exploring factors that predict smartphone and internet addiction may contribute to the behavioral treatment of these two of addiction behaviors.

### 1.1. Associations between Parental Psychological Control and Addiction Behaviors Surrounding Smartphones and the Internet

Parental psychological control, a specific aspect of parental control, can be regarded as a parental behavior that intrudes on and manipulates children’s thoughts, feelings, and attachments to their parents [7,8]. Such parenting practices can be assessed as parents’ psychologically manipulative and repressing practices, including guilt induction, love withdrawal, and authority assertion [9]. Guilt induction can be described as a parental behavior that pushes children to comply with their parents’ requests by inducing guilt in children [9]. Love withdrawal refers to parents withdrawing their love and care for children until the children’s performance meets their standards [10]. Authority assertion is a parental behavior in which parents inhibit children’s freedom to express their own feelings and opinions [10]. Parental psychological control is significantly different from parental behavioral control. The former type of parental control interferes with children’s psychological development by affecting their thoughts and emotions, while the latter attempts to control or manage children’s behavior [8,9].

Parental psychological control is considered an important factor associated with individuals’ addiction behaviors. Extensive studies have found that parental psychological control is positively associated with adolescents’ smartphone and internet addiction [11,12,13,14,15]. Three theories have been widely proposed to explain these relationships. The first theory (the self-determination theory) hypothesizes that self-determination either promotes or weakens adolescents’ intrinsic motivation and internalization, which affects their sense of autonomy, identity, and competence, resulting in excessive smartphone and internet use as a dysfunctional way of satisfying their psychological needs [16,17]. The second theory, which scholars label the social-bonding theory, assumes that parental psychological control is thought to be associated with children’s lower level of social bonding, which interferes with the development of an individual’s interpersonal relationships, and ultimately increased their levels of smartphone and internet addiction [11]. The third theory (the ecological systems theory) suggests parental psychological control influences children‘s professional needs [18]. Children show an imbalance in their mental state when their professional needs are inconsistent with their parents’ professional needs. In this case, these children might be at high-risk for these addiction behaviors.

Despite that both smartphone and internet addiction in adolescents are positively associated with parental psychological control, there is evidence that smartphone addiction differs significantly from internet addiction [3]. Previous studies have found that smartphone addiction is the excessive use of smartphones and internet addiction is the overuse of computers [3,19]. In addition, there are differences in personality traits, psychosocial factors, and gender between the two types of addictions [2,20,21,22]. Finally, evidence shows that smartphone addiction is more harmful than internet addiction [3,19].

### 1.2. Parental Psychological Control, Shyness, and Addiction Behaviors Surrounding Smartphones and the Internet

Shyness is an anxious preoccupation with imagined and real social situations [23,24]. There is evidence that parental psychological control has been found to be positively associated with child shyness in China [25]. Moreover, a study performed in Canada has observed that psychological control is related to greater peer exclusion in shy girls [26]. Finally, Finnish researchers have shown that psychological control can lead to shy children increasingly internalizing problems [27].

Increasing numbers of studies have observed that shyness can positively predict smartphone and internet addiction [28,29,30]. The online disinhibition effect indicated that online social exchange is more attractive for shy persons because it allows for invisibility and anonymity [31]. In this regard, online social exchange is a useful coping strategy for shy persons, resulting in increasing time on smartphones and the internet [32]. In addition, the theory of use and need can explain the associations between shyness and addiction behaviors surrounding the internet and smartphones [33].

### 1.3. The Present Study

The main goal of the present study was to investigate the associations between parental psychological control, shyness, and addiction behaviors surrounding smartphones and the internet. Based on the above literature review, we predicted that there were some differences between smartphone and internet addiction (e.g., sibling) (*Hypothesis 1*). Moreover, we hypothesized that three dimensions of parental psychological control were positively associated with addiction behaviors (*Hypothesis 2*). Finally, we expected that the relations between the three dimensions of parental psychological control (guilt induction, love withdrawal, authority assertion) and the two types of addiction behaviors were partially mediated by shyness (*Hypothesis 3*).

## 2. Methods

### 2.1. Participants and Procedures

Participants were adolescents (961 female, 896 male) who came from Chinese junior and senior middle schools. The participants were 11–18 years old (*Mean* = 14.93 years, *Standard Deviation* = 1.662 years). For mothers’ educational level, 246 mothers had a university or postgraduate degree, and 1262 mothers had a junior or senior middle school degree. For fathers’ educational level, 315 fathers had a university or postgraduate degree, 1274 fathers had a junior or senior middle school degree. In addition, 247 of the adolescents’ families showed low financial status (<4000 yuan per month; renminbi), 900 of the adolescents’ families showed medium financial status (4000–6000 yuan per month), and 653 of the adolescents’ families showed high financial status (>6000 yuan per month). Finally, 228 adolescents lived in one-parent families, and 1629 adolescents lived in two-parent families. In a 30 min class, adolescents were required to fill out four self-reported questionnaires. All subjects gave their informed consent for inclusion before they participated in the study. The study was conducted in accordance with the Declaration of Helsinki, and the protocol was approved by the Ethics Committee of College of Preschool and Primary Education in China West Normal University.

### 2.2. Measures

#### 2.2.1. Parental Psychological Control

The levels of parental psychological control were evaluated by the Parental Psychological Control Scale devised by Wang et al. [34]. The scale comprised 18 items, each of which was rated by using a 5-point scale ranging from 1 (not at all true) to 5 (very true). The 18 items were divided into 3 dimensions: guilt induction (10 items; e.g., “My parents tell me that I should feel guilty when I do not meet their expectations”), love withdrawal (5 items; e.g., “My parents act cold and unfriendly if I do something they do not like”), and authority assertion (3 items; e.g., “My parents tell me that what they want me to do is the best for me and I should not question it”). A higher score indicated a higher level of parental psychological control. The scale has been indicated to have good reliability and validity for determining parenting styles in Chinese populations. In our study, the Cronbach’s alpha coefficient for the Parental Psychological Control Scale was 0.937. The Cronbach’s alpha coefficients for the three subscales in the present study were 0.896 (guilt induction), 0.901 (love withdrawal), and 0.816 (authority assertion).

#### 2.2.2. Smartphone Addiction

Smartphone addiction was measured using the Mobile Phone Addiction Index [35]. The MPAI comprised 17 items, each of which was rated on a 5-point scale (1 = never, 5 = always). The scale included four aspects: inability to control cravings (7 items; e.g., “You have been told that you spend too much time on your smartphone”), anxiety and feeling lost (4 items; e.g., “You think it is hard to shut down your smartphone”), withdrawal and escape (3 items; e.g., “I have used my mobile phone to talk to others when I was feeling lonely”), and productivity loss (3 items; “Your productivity has decreased as a direct result of the time you spend on the smartphone”). A higher total score across all items indicated a greater likelihood of smartphone addiction. In this research, the Cronbach’s alpha coefficient was 0.892. For the four subscales, the Cronbach’s alpha coefficients were 0.805 (inability to control cravings), 0.751 (anxiety and feeling lost), 0.785 (withdrawal and escape), and 0.742 (productivity loss).

#### 2.2.3. Internet Addiction

Internet addiction was assessed with the Internet Addiction Test [36]. The scale is a 20-item questionnaire scored on a 5-point Likert scale ranging from 1 (very rarely) to 5 (very frequently). Sample items included “How often do you try to cut down the amount of time you spend online and fail?” and “Do you stay online longer than originally intended?”. The IAT has been demonstrated to have good reliability and validity in Chinese samples [37,38]. Higher scores represented higher levels of internet addiction. The Cronbach’s alpha in the present study was 0.914.

#### 2.2.4. Shyness

The adolescents’ shyness was measured with the Shyness Scale devised by Cheek and Buss [39]. The Chinese version of this scale was composed of 13 items (e.g., “I feel tense when I stay with people who are not familiar to me”). Each item was answered using a 5-point Likert-type scale ranging from 1 (not at all true) to 5 (very true). The shyness scale has been found to be reliable and valid in previous studies, including those set in China [28]. The Cronbach’s alpha in the present study was 0.845.

### 2.3. Analytic Strategy

First, descriptive and correlational analyses of all study variables were performed using IBM SPSS statistic 25. Next, structural equation modeling (SEM) was carried out to examine the mediating role of shyness in the relationship between the three dimensions of parental psychological control and the two types of addiction behaviors (smartphone and internet addiction). We used the Mplus 7 to evaluate the proposed models [40]. Missing data were processed using the full information maximum likelihood (FIML) procedure. Finally, the SEM was performed via the robust maximum likelihood (MLR) estimator to account for identified data non-normality, and the bootstrapping procedure was used to test indirect effects [41].

To evaluate the fit of the models to the data, we used the following standard fit indices [42]: a comparative fit index (*CFI*) > 0.90; a Tucker–Lewis index (*TLI*) > 0.90; a root-mean-square error of approximation (*RMSEA*) < 0.08; and a standardized root-mean-square residual (*SRMR*) < 0.10.

## 3. Results

### 3.1. Descriptive Statistics of the Study Variables

Table 1 summarized means, standard deviations, gender differences, and sibling differences with regard to parental psychological control, shyness, and smartphone and internet addiction. Results showed that females’ shyness and smartphone and internet addiction scores were significantly higher than males’ scores (shyness: *p* < 0.001; smartphone addiction: *p* < 0.001; internet addiction: *p* < 0.001). Although we failed to observe a sibling difference (Non-sibling vs. Sibling) in smartphone addiction (*p* = 0.267), adolescents with siblings showed higher levels of internet addiction than ones with no siblings (*p* = 0.037). The correlations between parental psychological control, shyness, smartphone addiction, internet addiction, and adolescents’ age and gender were listed in Table 2. The three dimensions of parental psychological control (guilt induction, love withdrawal, and authority assertion) were significantly positively correlated with shyness (*p* < 0.01), smartphone addiction (*p* < 0.01), and internet addiction (*p* < 0.01). Our further analyses showed that the correlation between authority assertion and smartphone addiction was larger than that between authority assertion and internet addiction (smartphone: *r* = 0.249; internet: *r* = 0.187; *u* = 2.00 > 1.96). Moreover, guilt induction was negatively correlated with age (*p* < 0.01). We also found that shyness was significantly positively associated with the following variables: smartphone addiction (*p* < 0.01), internet addiction (*p* < 0.01), and adolescents’ age and gender (*p* < 0.01). Finally, two forms of addiction behaviors were significantly positively associated with age and gender (*p* < 0.01).

### 3.2. Measurement Model

The present work performed separate confirmatory factor analyses (CFA) for the data of the three dimensions of parental psychological control to test whether these data fit the measurement models. The first measurement model included two types of variables: latent and observed variables. There were four latent variables (guilt induction, shyness, and smartphone and internet addiction) and thirteen observed variables. We observed a good data fit for the first measurement model: χ^2^ (55, *n* = 1857) = 644.377 (*p* < 0.001), χ^2^/df = 11.716; CFI = 0.961, TLI = 0.945; RMSEA = 0.076 (90% CI = 0.071–0.081); SRMR = 0.033. In addition, the second measurement model involved four latent variables (love withdrawal, shyness, and two forms of addiction behaviors) and fifteen observed variables. There was a good data fit for this model: χ^2^ (80, *n* = 1857) = 935.690 (*p* < 0.001), χ^2^/df = 11.696; CFI = 0.950, TLI = 0.934; RMSEA = 0.076 (90% CI = 0.072–0.080); SRMR = 0.036. Finally, the third model involved four latent variables (authority assertion, shyness, and addiction behaviors) and thirteen observed variables. A good data fit for this model was as follows: χ^2^ (55, *n* = 1857) = 659.193 (*p* < 0.001), χ^2^/df = 11.985; CFI = 0.954, TLI = 0.935; RMSEA = 0.077 (90% CI = 0.072–0.082); SRMR = 0.034. Each indicator of latent variables in the three measurement models was significant (*p* < 0.001), indicating that the indicators of latent factors could represent these latent factors

### 3.3. Structural Equation Modeling

#### 3.3.1. Guilt Induction, Shyness, and Addiction Behaviors

The model of the associations between guilt induction, shyness, and the two types of addiction behaviors (smartphone and internet addiction), while controlling for gender and age for the entire sample (see Figure 1), showed a good fit to the data, χ^2^ (59, *n* = 1857) = 644.377 (*p* < 0.001), χ^2^/df = 10.922; CFI = 0.961, TLI = 0.949; RMSEA = 0.073 (90% CI = 0.068–0.078); SRMR = 0.033. Our mediation analysis demonstrated that the relationship between guilt induction and the two types of addiction behaviors was partially mediated by shyness. The direct effects of guilt induction on these addiction behaviors were significant (smartphone: β = 0.236, SE = 0.028, *p* < 0.001, CI = 0.182–0.290; internet: β = 0.175, SE = 0.026, *p* < 0.001, CI = 0.125–0.225), and the indirect effects of guilt induction and both addiction behaviors mediated by shyness were significant (smartphone: β = 0.049, SE = 0.009, *p* < 0.001, CI = 0.031–0.068; internet: β = 0.053, SE = 0.010, *p* < 0.001, CI = 0.033–0.072; see Table 3).

#### 3.3.2. Love Withdrawal, Shyness, and Addiction Behaviors

The model examined the associations between love withdrawal, shyness, and the two forms of addiction behaviors, while controlling for gender and age for the entire sample (see Figure 2). The results showed a good fit to the data, χ^2^ (84, *n* = 1857) = 935.690 (*p* < 0.001), χ^2^/df = 11.139; CFI = 0.950, TLI = 0.937; RMSEA = 0.074 (90% CI = 0.070–0.078); SRMR = 0.036. Our mediation analysis demonstrated that the relationships between love withdrawal, shyness, and the two forms of addiction behaviors were partially mediated by shyness. The direct effects of love withdrawal on these addiction behaviors were significant (smartphone: β = 0.238, SE = 0.029, *p* < 0.001, CI = 0.181–0.295; internet: β = 0.203, SE = 0.028, *p* < 0.001, CI = 0.147–0.258), and the indirect effects of love withdrawal on these addiction behaviors mediated by shyness were significant (smartphone: β = 0.056, SE = 0.010, *p* < 0.001, CI = 0.037–0.075; internet: β = 0.059, SE = 0.010, *p* < 0.001, CI = 0.039–0.079; see Table 3).

#### 3.3.3. Authority Assertion, Shyness, and Addiction Behaviors

Finally, the model of the associations between authority assertion, shyness, and both types of addiction behaviors, after controlling for gender and age for the entire sample (see Figure 3), showed a good fit to the data, χ^2^ (59, *n* = 1857) = 659.193 (*p* < 0.001), χ^2^/df = 11.173; CFI = 0.955, TLI = 0.940; RMSEA = 0.074 (90% CI = 0.069–0.079); SRMR = 0.034. Our mediation analysis demonstrated that the relationships between authority assertion, shyness, and both types of addiction behaviors were partially mediated by shyness. The direct effects of authority assertion on these addiction behaviors were significant (smartphone: β = 0.242, SE = 0.028, *p* < 0.001, CI = 0.187–0.296; internet: β = 0.153, SE = 0.026, *p* < 0.001, CI = 0.103–0.204), and the indirect effects of authority assertion on these addiction behaviors via shyness were significant (smartphone: β = 0.039, SE = 0.010, *p* < 0.001, CI = 0.019–0.058; internet: β = 0.041, SE = 0.010, *p* < 0.001, CI = 0.021–0.061; see Table 3).

## 4. Discussion

The present study explored the relationship between parental psychological control (guilt induction, love withdrawal, and authority assertion), shyness, and addiction behaviors (smartphone and internet addiction). Our independent-sample *t*-tests showed that adolescents with siblings showed a higher level of internet addiction than those with no siblings in spite of a failure to observe a sibling difference with regard to smartphone addiction. Correlational analyses demonstrated that the three dimensions of parental psychological control were positively associated with adolescents’ addiction behaviors. The following analyses showed that the correlation between authority assertion and smartphone addiction was larger than that between authority assertion and internet addiction. Moreover, we observed that shyness was significantly positively related to the three dimensions of parental psychological control and two forms of addiction behaviors. Importantly, we found that the relations between the three dimensions of parental psychological control and adolescents’ addiction behaviors surrounding smartphones and the internet were partially mediated by shyness. Overall, our results indicate that shyness played an important role in the relationship between parental psychological control and addiction behaviors.

The present study observed that adolescents who had siblings showed higher levels of internet addiction than adolescents who had no siblings, which indicated the effect of siblings on adolescents’ internet addiction. Based on the interpersonal theory, parent–child relationships were negatively correlated with children’s internet addiction [43]. In particular, the adolescents who had siblings employed the internet as a compensatory tool to escape from the state of loneliness caused by the neglect of their parents, and this may lead to the development of an internet addiction [42,44]. However, we did not observe a sibling difference with regard to smartphone addiction. The differences between smartphone and internet addiction were similar to the findings of previous studies investigating these addiction behaviors. For instance, there were differences in psychosocial and personality traits between the two types of addictions [2,18,19,20].

The current study showed that the three forms of parental psychological control were positively associated with the two types of addiction behaviors. This finding was in accordance with previous studies examining the association between parental psychological control and the two types of addiction behaviors [45,46]. As our introduction demonstrated, this finding can be explained by two theories (the self-determination theory and social-bonding theory) [11,15,16]. Additionally, our further analyses showed that the correlation between authority assertion and smartphone addiction was larger than the correlation between authority assertion and internet addiction. This observation was consistent with previous studies investigating the differences between smartphone and internet addiction [2,21].

We found that the three dimensions of parental psychological control were significantly positively related to adolescents’ shyness, indicating that adolescents might show high levels of shyness when their parents exhibit stronger psychological control. This observation was consistent with a recent three-level meta-analysis, which showed that parental psychological control is positively related to youths’ internalizing problem behaviors [8]. Moreover, we observed that shyness was significantly positively related to addiction behaviors surrounding smartphones and the internet. This finding could be explained by the theory of use and need, which suggested that the use of smartphones and the internet could fulfill shy adolescents’ psychological needs [30].

A notable finding was that shyness mediated the relationships between the three dimensions of parental psychological control and adolescents’ addiction behaviors surrounding smartphones and the internet, which indicated that shyness was a vital link between these variables. It might be that parents’ high levels of psychological control led to adolescents’ high shyness, and ultimately these children showed severe smartphone and internet addiction [30,44,46]. Similarly to our finding, a previous study found that loneliness was an important intermediary factor in the association between parental psychological control and smartphone addiction [47]. Thus, it is likely that the relationships between these variables might be mediated by basic personal characteristics (e.g., shyness and loneliness).

Females’ shyness scores were significantly higher than males’ scores. This finding was documented by a number of previous studies examining people’s shyness [48,49,50]. In addition, females showed higher scores for internet addiction than males. This observation was inconsistent with previous studies examining people’s internet addiction, which showed no sex differences in the scores of internet addiction and higher scores for males for smartphone addiction [51,52]. One can speculate that these inconsistencies may be due to differences in the methods used to measure internet addiction (various scales of internet addiction).

## 5. Limitations

Like other studies, the present study was not without limitations. A cross-sectional method was employed to conduct our research, which failed to establish a causal relationship between the three dimensions of parental psychological control, shyness, and smartphone and internet addiction. Future scholars should address this limitation by using a longitudinal method. Moreover, the data were collected using a self-report format, generating misleading findings because it relied on the honesty and cooperation of participants. The data collection, in future studies, should combine self-report and partner-report items. Finally, our results cannot be generalized to other age groups (e.g., infants and adults) because the volunteers in the present study were adolescents.

## 6. Conclusions

The present study was a new attempt to explore the relationships between three dimensions of parental psychological control, shyness, and smartphone and internet addiction. Our results suggested that shyness played an important role in the associations between parental psychological control and the two forms of addiction behaviors. The results support the growing consensus that shyness and parental psychological control should be addressed when treating adolescents’ smartphone and internet addiction. Moreover, this study could provide a better understanding of the importance of parenting in adolescents’ psychological development. Given the limited number of studies examining parental psychological control, social withdrawal, and addiction behaviors, further studies are needed to investigate additional social withdrawal factors that might link shyness and smartphone and internet addiction.

## Figures and Tables

**Figure 1 ijerph-19-16702-f001:**
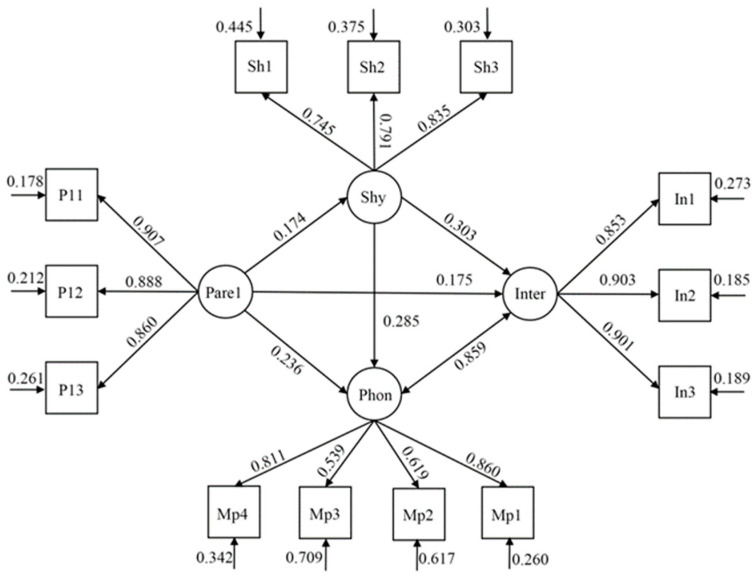
Structural equation model regarding the mediating effects of shyness on the association between guilt induction and adolescents’ two forms of addiction behaviors (smartphone and internet addiction). Note: Shy (shyness), Sh1–Sh3 three parcels of evaluative concerns regarding shyness; Pare1 (guilt induction), P11–P13 three parcels of evaluative concerns regarding guilt induction; Phon (smartphone addiction), MP1–MP4 four parcels of evaluative concerns regarding smartphone addiction; Inter (internet addiction), In1–In3 three parcels of evaluative concerns regarding internet addiction.

**Figure 2 ijerph-19-16702-f002:**
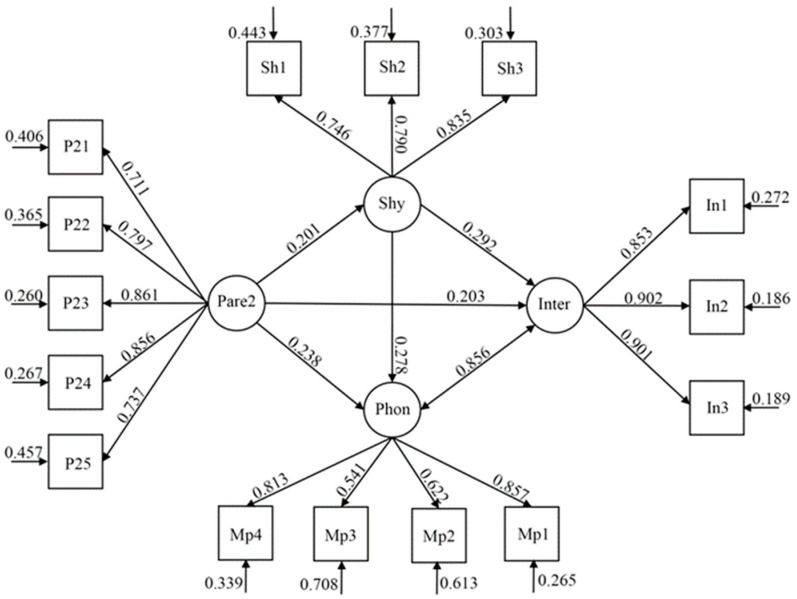
Structural equation model regarding the mediating effects of shyness on the association between love withdrawal and adolescents’ two forms of addiction behaviors. Note: Shy (shyness), Sh1–Sh3 three parcels of evaluative concerns shyness; Pare2 (love withdrawal), P21–P25 five parcels of evaluative concerns regarding love withdrawal; Phon (smartphone addiction), MP1–MP4 four parcels of evaluative concerns regarding smartphone addiction; Inter (internet addiction), In1–In3 three parcels of evaluative concerns regarding internet addiction.

**Figure 3 ijerph-19-16702-f003:**
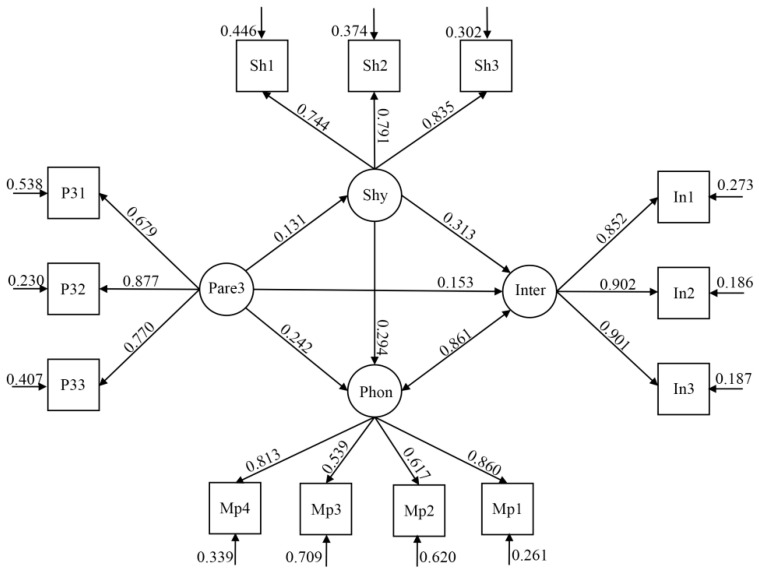
Structural equation model regarding the mediating effects of shyness on the association between authority assertion and adolescents’ two forms of addiction behaviors. Note: Shy (shyness), Sh1–Sh3 three parcels of evaluative concerns regarding shyness; Pare3 (authority assertion), P21–P25 three parcels of evaluative concerns regarding authority assertion; Phon (smartphone addiction), MP1–MP4 four parcels of evaluative concerns regarding smartphone addiction; Inter (internet addiction), In1–In3 three parcels of evaluative concerns regarding internet addiction.

**Table 1 ijerph-19-16702-t001:** Means, standard deviations, and gender effects of PPC, shyness, SA, and IA.

Variables	Gender Difference				Sibling Difference			
	Female	Male	*t*	*p*	Non-Sibling	Sibling	*t*	*p*
**Shyness**	38.90 ± 9.472	37.19 ± 9.837	−3.828	<0.001	37.33 ± 9.855	38.38 ± 9.604	−2.105	0.035
**PPC1**	24.77 ± 9.609	24.89 ± 8.501	0.294	0.769	25.48 ± 9.012	24.58 ± 9.112	1.923	0.939
**PPC2**	9.68 ± 5.079	9.72 ± 4.705	0.184	0.854	9.74 ± 4.955	9.68 ± 4.882	0.238	0.812
**PPC3**	8.96 ± 3.552	8.64 ± 3.397	−1.960	0.050	9.06 ± 3.562	8.71 ± 3.445	1.995	0.046
**SA**	42.19 ± 12.493	39.62 ± 13.033	−4.335	<0.001	40.43 ± 13.056	41.16 ± 12.720	−1.111	0.267
**IA**	50.25 ± 14.812	47.73 ± 15.736	−3.544	<0.001	47.86 ± 15.348	49.50 ± 15.277	−2.088	0.037

**Note**: PPC1: Guilt induction; PPC2: Love withdrawal; PPC3: Authority assertion; SA: Smartphone addiction; IA: Internet addiction; Non-sibling: A child in one family; Sibling: The number of children in one family was more than one.

**Table 2 ijerph-19-16702-t002:** Correlations between PPC, shyness, SA, and IA.

Variables	1	2	3	4	5	6	7	8
**1 PPC1**	—							
**2 PPC2**	0.718 **	—						
**3 PPC3**	0.687 **	0.548 **	—					
**4 Shyness**	0.154 **	0.180 **	0.128 **	—				
**5 SA**	0.264 **	0.281 **	0.249 **	0.270 **	—			
**6 IA**	0.216 **	0.245 **	0.187 **	0.290 **	0.752 **	—		
**7 Age**	−0.064 **	−0.006	−0.045	0.162 **	0.253 **	0.243 **	—	
**8 Gender**	−0.007	−0.004	0.045	0.089 **	0.100 **	0.082 **	0.038	—

**Note**: PPC1: Guilt induction; PPC2: Love withdrawal; PPC3: Authority assertion; SA: Smartphone addiction; IA: Internet addiction; and ** *p* < 0.01.

**Table 3 ijerph-19-16702-t003:** Standardized indirect effects of shyness on addiction behaviors in smartphone and internet.

Indirect Effect	β (Standardized Indirect Effect)	*SE*	*p*	95% CI Standardized Indirect Effect
From shyness to smartphone addiction				
PPC1:	(0.174) × (0.285) = 0.049	0.009	*p* < 0.001	0.031, 0.068
PPC2:	(0.201) × (0.278) = 0.056	0.010	*p* < 0.001	0.037, 0.075
PPC3:	(0.131) × (0.294) = 0.039	0.010	*p* < 0.001	0.019, 0.058
From shyness to internet addiction				
PPC1:	(0.174) × (0.303) = 0.053	0.010	*p* < 0.001	0.033, 0.072
PPC2:	(0.201) × (0.292) = 0.059	0.010	*p* < 0.001	0.039, 0.079
PPC3:	(0.131) × (0.313) = 0.041	0.010	*p* < 0.001	0.021, 0.061

**Note**: PPC1: Guilt induction; PPC2: Love withdrawal; PPC3: Authority assertion.

## Data Availability

Data will be provided if requested to the authors.

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
