# Peer review of "Parental Psychological Control and Addiction Behaviors in Smartphone and Internet: The Mediating Role of Shyness among Adolescents"

_ijerph, 2022, doi:10.3390/ijerph192416702_

Round 1

Reviewer 1 Report

The study used a positivist paradigm. But the paradigm was not declared in order to provide the language of the study especially under the issues of research methodology.

The study adequately identified the self-determination theory and the social-bonding theory that address individual and social needs respectively. The two theories only deal with the addictions or actions that have to do with societal and personal issues. What seems to be missing in the study is a theory that addresses a professional needs so that the results of the study can be framed to balance the solutions of societal, personal, and professional actions. 

Technically, words should be written in full. For example, is should be "...had not or had no sibling..." instead of "...hadn’t sibling..."

Author Response

Dear reviewer,        

Thank you for reviewing our manuscript (Title: Parental psychological control and addiction behaviors in smartphone and internet: The mediating role of shyness among adolescents; ID: ijerph-2064873). We appreciate the time and effort that you have taken to evaluate the work and find that all comments contribute to improving the paper. Below, we provided your comments, which followed by our responses and locations of changes in the paper. Words in blue are the changes made in the revised manuscript.

Thank you for the time and thoughts that you will have invested in assessing this revision.

Sincerely,

Qi Zhang

Responses to reviewer #1

Comment 1: The study used a positivist paradigm. But the paradigm was not declared in order to provide the language of the study especially under the issues of research methodology.

Response: Thanks for your review as well as this important feedback. We strongly agree with you. The paradigm has been declared in the revised Abstract. Please see page 1,Abstract” section, sentence 4-6.

Comment 2: The study adequately identified the self-determination theory and the social-bonding theory that address individual and social needs respectively. The two theories only deal with the addictions or actions that have to do with societal and personal issues. What seems to be missing in the study is a theory that addresses the professional needs so that the results of the study can be framed to balance the solutions of societal, personal, and professional actions. 

Response: We find this comment to be insightful. We have provided a theory that addressed the professional needs in the revised manuscript. Please see page 2, section1.1, paragraph 2, last three sentences.

Comment 3: Technically, words should be written in full. For example, is should be "...had not or had no sibling..." instead of "...hadn’t sibling..."

Response: Thanks for your review and bringing it to our attention. The words have been written in full in the revised manuscript. In addition, our paper has been edited by a native speaker.

Reviewer 2 Report

This is an interesting paper with valuable insight into a much-needed area of research.   

Abstract 
The abstract introduces the content of the paper appropriately. 

Introduction 
The introduced theories and previous research results are in line with the content and direction of the paper.
In Section 1.2, it is worth mentioning research results globally (outside China)
The aims and hypotheses introduced are appropriate and fit the context.

Methods
The Participants subchapter contains important information, however, some relevant and crucial information are missing. It would be necessary to introduce the background of the participants (e.g. parents' educational level, financial status, family environment) if available.
The introduction of the measures is correct as well. However, the Cronbach alpha of the subscales can still be added.

Results​
Results are introduced clear. â€‹The division of the chapters is logic, the introduction of the results of the analyses is correct. The tables and figures help the reader to understand the results.
 â€‹
Discussion​
Discussion is correctly defined. The authors reflected well on previous findings and theories. This makes the paper coherent.

Limitations
Limitations mentioned are correct. It should be also emphasised that results cannot be generalised to the population while the sample was not representative.

Conclusions
This section should be extended with the practical relevance of the paper. How can it be used in diverse contexts (psychological practice, school, parenting etc.).

Overall, this paper is a valueable study which is worth publishing after minor modifications.

Author Response

Dear reviewer,        

Thank you for reviewing our manuscript (Title: Parental psychological control and addiction behaviors in smartphone and internet: The mediating role of shyness among adolescents; ID: ijerph-2064873). We appreciate the time and effort that you have taken to evaluate the work and find that all comments contribute to improving the paper. Below, we provided your comments, which followed by our responses and locations of changes in the paper. Words in blue are the changes made in the revised manuscript.

Thank you for the time and thoughts that you will have invested in assessing this revision.

Sincerely,

Qi Zhang

Responses to reviewer #2

Comment 1: This is an interesting paper with valuable insight into a much-needed area of research.   

Response: Thanks for your review as well as this important feedback.

Comment 2: The abstract introduces the content of the paper appropriately.

Response: Thank you.

Comment 3: The introduced theories and previous research results are in line with the content and direction of the paper. In Section 1.2, it is worth mentioning research results globally (outside China). The aims and hypotheses introduced are appropriate and fit the context.

Response: Thank you again.

Comment 4: The Participants subchapter contains important information, however, some relevant and crucial information are missing. It would be necessary to introduce the background of the participants (e.g. parents' educational level, financial status, family environment) if available. The introduction of the measures is correct as well. However, the Cronbach alpha of the subscales can still be added.

Response: We are thankful for this suggestion. We have introduced the background of the participants (e.g. parents’ educational level, financial status, family environment) in the revised manuscript. Please see page 3, section2.1, paragraph 1, blue words.

Moreover, the Cronbach alphas of subscales for the Parental Psychological Control Scale and Mobile Phone Addiction Index have be added in the revised section 2.2.1 and 2.2.2, blue words.

Comment 5: Results are introduced clear. â€‹The division of the chapters is logic, the introduction of the results of the analyses is correct. The tables and figures help the reader to understand the results.

Response: Thanks for your review as well as this important feedback.

Comment 6: Discussion is correctly defined. The authors reflected well on previous findings and theories. This makes the paper coherent.

Response: Thank you.

Comment 7: Limitations mentioned are correct. It should be also emphasised that results cannot be generalised to the population while the sample was not representative.

Response: Thank you for your time and careful attention to our manuscript. We have emphasised that results cannot be generalised to the population in the revised limitations, blue words.

Comment 8: This section should be extended with the practical relevance of the paper. How can it be used in diverse contexts (psychological practice, school, parenting etc.). Overall, this paper is a valueable study which is worth publishing after minor modifications.

Response: Thanks for your review as well as this important feedback. We have extended with the practical relevance of the paper. Please see page 11, section6, blue words.

Reviewer 3 Report

Very interesting study!

Introduction:

- the readers may benefit from explaining more what "parental psychological control" means.  On line 89-90, some examples are provided

- literature review needs more context. Great job in providing the theoretical orientation and citing a few articles, but since the authors are connecting multiple different experiences, more literature review regarding each of these constructs and their connection will strengthen the proposed hypothesis.

Methods:

- Line 95: "recruited from" rather than "were from" may be a more appropriate phrase.

- Table 1 looks messy on my end

Results:

- sibling finding was not contextualized and thus confusing. What is the hypothesis? Why do we believe having a sibling makes an impact. In discussion section, this result was not further explained either.

Author Response

Dear reviewer,        

Thank you for reviewing our manuscript (Title: Parental psychological control and addiction behaviors in smartphone and internet: The mediating role of shyness among adolescents; ID: ijerph-2064873). We appreciate the time and effort that you have taken to evaluate the work and find that all comments contribute to improving the paper. Below, we provided your comments, which followed by our responses and locations of changes in the paper. Words in blue are the changes made in the revised manuscript.

Thank you for the time and thoughts that you will have invested in assessing this revision.

Sincerely,

Qi Zhang

Responses to reviewer #3

Comment 1: The readers may benefit from explaining more what “parental psychological control” means.  On line 89-90, some examples are provided.

Response: Thanks for your review as well as this important feedback. We strongly agree with you. We have explained more what “parental psychological control” means. Please see page 2, section1.1, paragraph 1, blue words.

Comment 2: Literature review needs more context. Great job in providing the theoretical orientation and citing a few articles, but since the authors are connecting multiple different experiences, more literature review regarding each of these constructs and their connection will strengthen the proposed hypothesis.

Response: Thank you for your time and careful attention to our manuscript. We have tried our best to provide more literature review regarding each of these constructs. Please see page 2, section1.1, paragraph 1, blue words.

Comment 3: Line 95: “recruited from” rather than "were from" may be a more appropriate phrase.

Response: We appreciate your guidance here. We have revised the error. Please see page 3, section2.1, sentence 1.

Comment 4: Table 1 looks messy on my end.

Response: Thank you for your time and careful attention to our manuscript. We have revised Table 1. Please see revised Table 1.

Comment 4: Sibling finding was not contextualized and thus confusing. What is the hypothesis? Why do we believe having a sibling makes an impact. In discussion section, this result was not further explained either.

Response: Thanks for your review as well as this important feedback. We agree with you strongly. Firstly, we have re-written the hypothesis on sibling. Please see page 3, section1.3, blue words. Secondly, we have explained the results of sibling finding. Please see page 9, section4, paragraph 2.